# Application of Texture and Fractal Dimension Analysis to Estimate Effectiveness of Oral Leukoplakia Treatment Using an Er:YAG Laser—A Prospective Study

**DOI:** 10.3390/ma13163614

**Published:** 2020-08-15

**Authors:** Kamil Jurczyszyn, Marcin Kozakiewicz

**Affiliations:** 1Department of Oral Surgery, Wroclaw Medical University, Krakowska 26, 50-425 Wroclaw, Poland; 2Department of Maxillofacial Surgery, Medical University of Lodz, 1stGen. J. Hallera Pl., 90-647 Lodz, Poland; marcin.kozakiewicz@umed.lodz.pl

**Keywords:** leukoplakia, texture analysis, fractal dimension analysis, oral pathology, laser surgery, Er:YAG laser

## Abstract

Background: Oral leukoplakia (OL) is a potential neoplasmic lesion. The aim of this study was to apply texture analysis (TA) and fractal dimension analysis (FDA) to estimate the effectiveness of OL treatment using an Er:YAG laser. Methods: Eighteen patients with 32 lesions were treated. Laser procedures were conducted using the LiteTouch™ Er:YAG Dental Laser. The diameter of the operational tip was 1.3 mm, the power was 50 mJ, the frequency was 50 Hz, and the wavelength was 2940 nm. TA was based on long and short-run emphasis inverse moments, difference entropy, inverse difference moment, and wavelet decomposition for two-dimensional photography. FDA was measured using the box-counting method. Results: Total response was achieved in 50% of lesions, partial response was observed in 47%, and 3% of lesions did not respond to treatment. Recurrence occurred in 34% of lesions. TA features indicated pathological images depicting leukoplakia and complete reconstruction of the correct mucosal image after laser ablation. The discrete wavelet transformation feature detects much larger structures than the properties derived from the run-length matrix and co-occurrence matrix. Conclusions: The Er:YAG laser is an effective treatment method in cases of oral leukoplakia. Leukoplakia treatment by Er:YAG laser is an effective modality, as revealed by the oral mucosa microstructure. TA and FDA are promising methods to estimate the effectiveness of OL treatment.

## 1. Introduction

Oral leukoplakia (OL) is a potential neoplasmic lesion [1]. OL is a white patch or plaque that cannot be clinically identified as any other disease. The classical definition of leukoplakia is presented by the World Health Organisation. Malignant transformation risk is between 0.2 and 9% [2]. The rate of malignant transformation depends on the follow-up time and demographic factors [3]. Leukoplakia may be divided into homogenous and nonhomogeneous types. Nonhomogeneous OL presents a higher risk for malignant transformation [4]. Warnakulasuriya and Ariyawardana revealed a total transformation rate of 14.5% for nonhomogeneous leukoplakia [5]. Cigarette smoking, alcohol, spicy food, galvanic current, and mechanical irritation are the factors most commonly affecting leukoplakia. Classical surgery is still helpful in the case of small lesions. Widespread lesions that involve the whole mucous membrane of the cheeks, tongue, bottom of the oral cavity, or alveolar ridge are important clinical problems. In such cases, classical surgery fails. Classical excision of mucosa on a large surface area, without reconstructive treatments, will lead to scar formation, impairing the function of the stomatognathic system. Cryosurgery, photodynamic therapy, systemic administration of vitamin D, and laser surgery are known as less invasive treatment methods of OL.

In our study, we used an Er:YAG laser. The Er:YAG laser emits light with a 2940 nm wavelength, which corresponds to the main peak of water absorption [6]. Most of the energy is absorbed in the epidermis and papillary dermis. This laser results in more superficial ablation with less thermal damage [7].

Clinical estimate of treatment effectiveness is a problem due to the irregular shape of lesions. In the case of the tongue and cheeks, the shape of lesions depends on the mucous membrane tension. In small lesions, when regression is completed, diagnosis is simple: clinical examination and, alternatively, excision for controlled microscopic examination. Large, multifocal lesions are very often only partially cured. It is difficult to estimate whether a cured region looks like a normal mucous membrane only by performing macroscopic examination without microscopic examination as well. In our study, we tried to use texture analysis and fractal dimension analysis to achieve more objective methods for examination during and after treatment.

Digital images consist of pixels. Pixels create the delicate structure of an image, which is named texture. Texture is a collection of recurrent graphical patterns characterized by brightness, entropy, smoothness, uniformity, roughness, granulation, randomness, or linearity [8]. Mathematical and statistical analysis of texture patterns is known as texture analysis (TA). TA is based upon the mathematical analysis of the matrix that represents the distribution of pixel brightness in the image area. Texture analysis may be divided into four methods: statistical, structural, model-based, and transform [9,10]. TA is commonly used in medicine to analyse X-ray photos, computed tomography, or magnetic resonance images [11,12,13,14]

Irregular shape is a characteristic feature of leukoplakia. The measurement of a leukoplasm’s area and shape is very difficult, especially comparing measurements before and after treatment. Fractal dimension analysis (FDA) is a useful method in cases with such complicated patterns. FDA is used when classical Euclidian mathematics fails.

Simple mathematic formulas describe the fractal, but these formulas are iterated up to infinity. Thanks to these iterations, fractals can be magnified up to infinity. On each level of magnification, we are able to see new details of fractals, but these features are similar to the origin shape. This feature is named a self-similarity.

Using classic Euclidean geometry on a daily basis, we are used to a line having only one dimension, which is its length; an item on a plane has two dimensions: length and width; while a three-dimensional item has height, width and length. However, fractals are shapes beyond the principles of classic geometry. Another surprising feature of fractals is their self-similarity. It is a feature manifested in the fact that a fractal can be magnified unlimitedly, and subsequent details of its structure are similar to its initial shape.

Figure 1 shows examples of fractals and their fractal dimensions (FD). The fractal dimension value of Cantor’s set is approximately 0.631, so it is something between a point (FD = 0) and a straight line (FD = 1). Koch’s snowflake (FD ≈ 1.262) is a shape closer to the line than a two-dimensional object, but Sierpinski’s triangle (FD ≈ 1.585) is almost in the centre between a line and a plane figure (FD = 2).

Many anatomical structures, such as blood vessels, neural networks, and bone structure, are examples of fractals.

The aim of this study was to apply texture and fractal dimension analysis to estimation the effectiveness of oral leukoplakia treatment using an Er:YAG laser.

## 2. Materials and Methods

### 2.1. Patients and Lesions

Eighteen patients were enrolled into the study (12 women (67%) and 6 men (33%)). The median age was 61 years (SD = 11). The youngest patient was 36, and the eldest patient was 83 years old. The median age was 65 years for women and 54 years for men. The overall number of lesions was 32. Exclusion criteria were dysplasia occurrence in histopathological examination and patients who smoke cigarettes.

After local anaesthesia, a specimen of the lesion was taken out for histopathological examination. The excision site was chosen on the border between healthy mucosa and the lesion. In the case of widespread lesions, two or three samples were collected from the most representative sites. Histopathological slides were stained with haematoxylin and eosin. The most common occurrence of leukoplakia was mucous of the cheek (11 lesions), tongue (6 lesions), and mucous membrane of the alveolar ridge (8 lesions). The least common occurrence of leukoplakia foci was in the palate region (only 2 lesions).

Lesions were categorized according to the van der Waal classification. Three parameters are the basis of that classification: L: lesion size (1: smaller than or equals 2 cm; 2: between 2 and 4 cm; 3: lesion larger than 4 cm); C: clinical view (1: homogenous; 2: nonhomogeneous); P: histopathologic (1: without dysplasia; 2: dysplasia present).

A decrease in at least one physical dimension of a lesion of up to 50% was a criterion of partial remission. All treatment procedures were conducted every two weeks. Control examinations were performed two weeks after the last laser procedure and at 1 month, 3 months, 6 months, and 1 year. Patients were informed to monitor the mucous membrane and call for a control visit if a suspicious lesion appeared.

Texture and fractal dimension analysis were performed in the following groups: reference (healthy oral mucosa), pre-operational (lesion), intermediate (lesion two weeks after treatment), and post-operational (region totally cured in clinical examination).

The study was done after obtaining an approval of the Ethics Committee of the Medical University in Wroclaw (approval number: KB -367/2014).

### 2.2. Laser Procedure

All procedures were performed with local infiltration anaesthesia using 1.8 mL of Septanest 200 (4% articaine with 1:200000 adrenaline, Septodont, Poland). Laser procedures were conducted using a LiteTouch™ Er:YAG Dental Laser (Light Instruments Ltd, Yokneam, Israel). The diameter of the operational tip was 1.3 mm, the power was 50 mJ, the frequency was 50 Hz, and the wavelength was 2940 nm; water cooling was applied. The total dose of radiation depended on the size of the lesion. All laser procedures were limited to the epithelial layer of the oral mucosa to decrease the risk of scar formation.

### 2.3. Image Acquisition

All photos were taken using a Samsung S9 (Samsung, Seoul, South Korea) cell phone camera. A cell phone camera was used, because in a future study, we will try to create a system of remote diagnosis of oral mucosa lesions on the basis of cell phone photography to simplify the whole process on the dentist’s side. All photos were taken perpendicular to the mucous membrane. To achieve the same scale of photos, a focus point was locked at the same distance. Images were stored at a maximum resolution (12 megapixels).

### 2.4. Image Preparation

All graphical operations were performed in GNU Image Manipulation Program (GIMP) version 2.10.18 (open source licence, www.gimp.org). In the centre of the lesion, a square region of interest (ROI, 150 × 150 pixels) was selected. The ROI was cut off from the original photo. We used the auto-levels tool to equalize the histogram of the image. Then, greyscale conversion was applied (prepared and saved images were put through texture analysis). After that, images were converted to bitmap format (threshold level 137 of 255). The threshold was set experimentally. Finally, images were stored in TIFF (tagged image file format) format (without compression). All graphical operations are shown in Figure 2. Such prepared images were the basis for counting the fractal dimension. All image processing was necessary to obtain source material for further analysis. The auto-levels tool enables stretching of a histogram, which increases contrast between the lesion and background. Greyscale conversion is needed to create bitmap files. The classical counting box method (fractal dimension analysis) requires bitmap format (one bit notation–1–signal, 0–background). The threshold of this conversion (134) was experimentally chosen to achieve sharp borders of lesions, and a threshold level of 128 (default binary conversion) blurred these borders. Texture analysis requires greyscale conversion.

### 2.5. Texture Analysis

Due to the determination of the usefulness of the run-length matrix as well as the co-occurrence matrix in digital imaging diagnosis of oral mucosal white lesions, a similar approach was used in this study to evaluate the results of laser treatment of oral leukoplakia [15]. Let *p*(*i*,*j*) be the number of times there is a run of length *j* having grey level *i*. Let *N_g_* be the number of grey levels and *N_r_* be the number of runs [16]. Definitions of the parameters of the run-length matrix *p*(*i*,*j*) are given below.

Long-run emphasis inverse moments: (1)$$LngREmph=\left(\overset{N_{g}}{\underset{i=1}{\sum }}\overset{N_{r}}{\underset{j=1}{\sum }}j^{2}p\left(i,j\right)\right)/C$$

Short-run emphasis inverse moments: (2)$$ShrtREmph=\left(\overset{N_{g}}{\underset{i=1}{\sum }}\overset{N_{r}}{\underset{j=1}{\sum }}\frac{p\left(i,j\right)}{j^{2}}\right)/C$$
where the coefficient *C* is defined as
(3)$$C=\overset{N_{g}}{\underset{i=1}{\sum }}\overset{N_{r}}{\underset{j=1}{\sum }}p\left(i,j\right)$$

The second-order histogram is defined as the co-occurrence matrix *h_dθ_* (*i*,*j*) [17]. When divided by the total number of neighbouring pixels *R* (*d*,*θ*) in the ROI, this matrix becomes the estimate of the joint probability, *p_dθ_* (*i*,*j*), of two pixels a distance *d* apart along a given direction *θ*, having particular (co-occurring) values *i* and *j*. Formally, given the image *f* (*x*,*y*) with a set of *N_g_* discrete intensity levels, the matrix *h_dθ_* (*i*,*j*) is defined such that its (*i*,*j*)th entry is equal to the number of times that
(4)$$f\left(x_{1},\;y_{1}\right)=i\;\mathrm{and}\;f\left(x_{2},\;y_{2}\right)=j$$
where $\left(x_{2},\;y_{2}\right)=\left(x_{1},\;y_{1}\right)+\left(d\;\cos\theta ,\;d\;\sin\theta \right)$.

This yields a square matrix of dimension equal to the number of intensity levels in the image, for each distance *d* = 5 pixels and orientation with angles *θ* = 0°, 45°, 90°, and 135° (these are considered, and next their average is calculated). Reduction in the number of intensity levels (by quantization to fewer levels of intensity) helps remove noise, with some loss of textural information (as low as 4-bit here). The co-occurrence matrix-derived parameters are defined by the equations that follow, where *p_x_* (*i*) and *p_y_* (*j*) are the marginal distributions.

### 2.6. Difference Entropy

(5)$$DifEntrp=-\overset{N_{g}}{\underset{i=1}{\sum }}p_{x-y}\left(i\right)\log\left(p_{x-y}\left(i\right)\right)$$

Inverse difference moment:(6)$$InvDfMom=\overset{N_{g}}{\underset{i=1}{\sum }}\overset{N_{g}}{\underset{j=1}{\sum }}\frac{1}{1+{\left(i-j\right)}^{2}}p\left(i,j\right)$$

Calculations were performed in Mazda 4.6 (Technical University of Lodz, Lodz, Poland) on selected features [15,18,19,20,21,22].

However, the above texture characteristics based on the run-length matrix and co-occurrence matrix detect quite small parts of the mucous membrane image. To evaluate objects visualized more globally, discrete wavelet transform is used [23,24,25]. To compute the wavelet features in the first step, the Harr wavelet is calculated for the whole image. The discrete wavelet transform (DWT) is a linear transformation that operates on a data vector whose length is an integer power of two, transforming it into a numerically different vector of the same length. It is a tool that separates data into different frequency components and then studies each component with resolution matched to its scale (scale 5 indicates bigger texture elements than scale 3). The discrete wavelet transform is computed with a cascade of filters followed by a factor 2 subsampling (Figure 3).

H and L denote high and low-pass filters, respectively (H detects lower objects; L detects higher objects). Boxes after each filter denote subsampling. Outputs of these filters are given by the following equations:(7)$$a_{j+1}\left[p\right]=\sum_{n=-\infty }^{+\infty }l\left[n-2p\right]d_{j}\left[n\right]$$
(8)$$d_{j=1}\left[p\right]=\sum_{n=-\infty }^{+\infty }h\left[n-2p\right]a_{j}\left[n\right]$$

Elements *a_j_* are used for the next step (scale) of the transform, and elements *d_j_*, called wavelet coefficients, determine the output of the transform. *l* [*n*] and *h* [*n*] are coefficients of low and high-pass filters, respectively. One can assume that on scale *j* + 1, there is only half of the number of *a* and *d* elements at scale *j*. This means that DWT can be done until only two *a_j_* elements remain in the analysed signal. These elements are called scaling function coefficients.

The discrete wavelet transform algorithm for pictures is similar. The discrete wavelet transform is performed first for all image rows and then for all columns. As a result of this transform there are 4 subband images at each scale (Figure 4).

The increase in energy calculated for the wavelet coefficient LH (higher objects) and HL (high objects but lower than detected in LH) indicates the detection of big longitudinal or oval image elements (Figure 5). Scale 5 indicates relatively big texture elements observed in the image too [26]. The most effective detection of such two-dimensional objects has been achieved in this study for wavelet energy calculated after a high-pass filter (H) and next through a low-pass filter (L) in scale 5 (s5), according to the equation below:(9)$$WavEnHL\_s\text{-5}=\frac{\sum_{x,y\in ROI}{\left(d_{x,y}^{HL}\right)}^{2}}{n}$$
where *n* is the number of pixels in the ROI, both at the given scale and subband; *x*,*y* are the coordinates of a pixel; *d* is the wavelet coefficient.

Obviously, ROIs are reduced in size in successive scales in order to correspond to subband image dimensions. In a given scale, the energy is calculated only if the ROI at this scale contains at least 4 pixels. The output of this procedure is a vector of features containing energies of wavelet coefficients calculated in subbands at successive scales.

### 2.7. Fractal Dimension Analysis

Fractal dimension (*D_S_*) is counted using the formula below [27]:(10)$$D_{S}=\underset{\varepsilon \to 0}{\lim}\frac{\log N\left(\varepsilon \right)}{\log\left(\frac{1}{\varepsilon }\right)}$$
where *D_S_*—fractal dimension; ε—length of box that creates a mesh covering surface with the examining pattern; *N*(ε)—minimal number of boxes required to cover the examining pattern.

Graphical interpretation of the formula above involves data being applied on a chart, where the x-axis is a decimal logarithm of the reverse length of the grid side covering the image at each stage, while the y-axis is a decimal logarithm of the minimum number of grids needed to cover the studied shape at the same stage. A straight line goes through points set in such a way. The straight line is described by the formula: *y* = *ax* + *b*, and interpretation of the fractal dimension is calculating the value of the directional factor and of the straight line (Figure 6) [28]. All calculations of the fractal dimension were carried out using the Fractalyse 2.4 program (University of Franche-Comté, Besançon, France, www.fractalyse.org). Fractalyse software measures fractal dimension using the box-counting method.

### 2.8. Statistical Analysis

Normality was checked by Shapiro–Wilk test application. One-way analysis of variance was used for the detection of differences in the lesion, margin, and healthy tissue. The Kruskal–Wallis test was applied due to the presence of non-normal data distributions. Factor analysis was applied to obtain a small number of factors that account for most of the variability in the 4 variables (FDA, WavEnHL_s-4, DifEntr/LngREmph, ShrtREmp), which were later used for leukoplakia recurrence evaluation. A difference was considered significant if *p* < 0.05. Stargraphics Centurion 18 ver.18.1.12 (StarPoint Technologies, Inc., The Plains, VA, USA) and Statistica ver.13.3 (StatSoft, Krakow, Poland) were used for statistical analyses.

## 3. Results

Eighteen patients with 32 lesions were treated (1.78 lesion per patient). In total, 123 laser procedures were conducted (3.8 procedures per patient and 6.8 procedures per lesion). A total response was achieved in 16 patients (50%), a partial response was observed in 15 patients (47%) and one patient (3%) did not respond to treatment (despite six treatment procedure applications). In this case, the lesion was incised using the classical surgical method and sent for microscopic examination. Recurrence was noted in the case of 11 lesions (34%). In the case of recurrence, a new specimen was taken out for microscopic examination.

### 3.1. Texture Analysis

The development of leukoplakia causes characteristic changes in images of the oral cavity mucosa (Figure 5, Table 1). The incidence of long bright chains of pixels increases (*p* < 0.001). Laser ablation removes these structures in significant amounts, but there are still more of them than in healthy oral mucosa (*p* < 0.001). Only after a period of complete healing is the image of the mucous membrane at the site of the removed leukoplakia identical with LngREmp (Figure 7). ShrtREmp is the ideological opposite of LngREmph (Figure 7), but the transformations described by this feature are the same. They also prove the statistically significant efficacy of the Er:YAG laser in the treatment of leukoplakia (*p* < 0.001).

DifEntrp is a less susceptible (*p* < 0.05) feature when analysing photographic images of oral mucosa than features derived from the run-length matrix. However, DifEntrp indicates a decrease in the variation in pathology of the fine pattern found in healthy mucosa. This fine texture network is reproduced after the healing period (post-op, Figure 7). The last feature examined from the group originating from the co-occurrence matrix is InvDfMom, which again strongly describes pathological changes and mucosal healing (*p* < 0.001). The number of small, white fields increases in leukoplakia, and after full healing, the number of reticular (contrary to plate) patterns increases, which manifests itself as a decrease in InvDfMom values to the size characteristic of healthy mucosa.

All of the features described above indicate a pathological image presented by leukoplakia and complete reconstruction of the correct mucosal image after laser ablation. This applies to the microscale, but if one considers the macroscale, new information is provided by the assessment of the discrete wavelet transformation feature (Figure 8). There is a progressive increase in the energy of the HL subband in the direction of the reference images by leukoplakia (pre-operational), from the intermediate period up to post-operational (*p* < 0.001). This is related to the creation of extensive oval structures filling up most of the image samples. This feature detects much larger structures than features derived from the run-length matrix and co-occurrence matrix (Table 1).

### 3.2. Fractal Dimension Analysis

The lowest value of FDA was observed in the pre-operational group (1.810 ± 0.156), and the highest value was seen in the reference group (1.924 ± 0.119). FDA at a level of 1.923 ± 0.059 was noted in the intermediate period group, and FDA value 1.915 ± 0.082 was observed in the post-operational group. It is important that significant differences were seen in the case of the pre-operational group versus the reference group (*p* < 0.05), but no significant differences were seen between the reference, intermediate period, and post-operational groups. The statistical chart is shown in Figure 9, and the results of Kruskal–Wallis one-way analysis of variance are shown in Table 2.

### 3.3. Estimation of Recurrence

The purpose of the factor analysis was to obtain a small number of factors that accounted for most of the variability in the four selected variables. In this case, two factors were extracted, since both factors had eigenvalues greater than or equal to 1.0. Together, they accounted for 82% of the variability in the original data (Table 3).

Next, matrix rotation was performed for the easy and obvious naming of the calculated factor (Table 4).

This table shows the equations that estimate the common factors after rotation has been performed. The rotation was performed in order to simplify the explanation of the factors. The rotated factors have the following equations:FDT factor = 0.114858*FDA + 0.167196*WavEnHL_s-4 +0.971237*DifEntr/LngREmph + 0.976915*ShrtREmp(11)
FDA–WDT factor = 0.813578*FDA + 0.784505*WavEnHL_s-4 +0.190267*DifEntr/LngREmph + 0.152573*ShrtREmp(12)
where the values of the variables in the equation are standardized by subtracting their means and dividing by their standard deviations. It also shows the estimated communalities, which can be interpreted as estimating the proportion of the variability in each variable attributable to the extracted factors. Both factors are oriented towards describing the state of the normal mucous membrane, because they were calculated from data for the reference oral mucosa. This means that such relationships are present in a healthy place, and the treatment should aim at achieving this mucosal condition (Figure 10 and Figure 11).

If an accumulation of locations with a similar tendency to recurrence is made, it is noted that recurrence of OL is more common after cheek and tongue treatment (42%). However, recurrences are less frequent (4.2%) when the primary lesion is treated within the gum or oral floor (Chi^2^ test 4.934, *p* < 0.05). The number of performed laser sessions did not affect the frequency of recurrence (ANOVA, F = 0.73, *p* = 0.401).

Appearance characteristics of the OL focus (second order features: FD and TA or third order features: FDT and FDA–WDT factors) have some prognostic significance (Table 5): pre-op FDA (recovery 1.73 ± 0.19 vs. recurrence 1.89 ± 0.07, *p* < 0.05), pre-op DifEntr/LngREmph (recovery 0.10 ± 0.06 vs. recurrence 0.06 ± 0.04, *p* = 0.054), pre-op FDT factor (recovery 1.00 ± 0.14 vs. recurrence 0.94 ± 0.08, *p* = 0.434), and FDA–WDT factor (recovery 2.03 ± 0.32 vs. recurrence 2.14 ± 0.21, *p* = 0.344). Monitoring of the progress of treatment (intermediate test period) of the above features has no prognostic value: intermediate FDA (recovery 1.92 ± 0.06 vs. recurrence 1.92 ± 0.07, *p* = 0.946), intermediate DifEntr/LngREmph (recovery 0.11 ± 0.06 vs. recurrence 0.11 ± 0.09, *p* = 0.664), intermediate FDT factor (recovery 1.05 ± 0.12 vs. recurrence 1.03 ± 0.19, *p* = 0.543), and intermediate FDA–WDT factor (recovery 2.16 ± 0.20 vs. recurrence 2.11 ± 0.20, *p* = 0.612). The sites after the treatment do not allow information about future potential relapse to be obtained: post-op/op FDA (recovery 1.92 ± 0.07 vs. recurrence 1.93 ± 0.09, *p* = 0.131), post-op DifEntr/LngREmph (recovery 0.15 ± 0.08 vs. recurrence 0.16 ± 0.05, *p* = 0.674), post-op FDT factor (recovery 1.10 ± 0.16 vs. recurrence 1.13 ± 0.08, *p* = 0.503), and post-op FDA–WDT factor (recovery 1.98 ± 0.16 vs. recurrence 2.05 ± 0.14, *p* = 0.289).

## 4. Discussion

Leukoplakia, especially the nonhomogeneous type, is a clinical problem as it is a potentially premalignant lesion. Risk of malignant transformation requires the application of effective treatment methods. Laser surgery is one such treatment. Matulić et al. used two lasers, Er:YAG and Er,Cr:YSGG, for leukoplakia treatment. Both Er:YAG and Er,Cr:YSGG lasers are efficient in the removal of oral leukoplakia without significant intraoperative or post-operative adverse effects [29]. Romeo et al. show that a previous smoking habit has an influence on treatment success. Patients with no history of smoking habits showed complete healing of 87.5%, while in ex-smokers, complete healing was 42.8%. The recurrence rate in the group without extension of margins was 45.5%. Achieving 3 mm margins during ablation reduces the recurrence rate to 36.4%, so the recurrence rate was similar to our study (34%) [30]. A similar recurrence level was reported by Galletta et al., in the case of CO_2_ laser applied in leukoplakia treatment. The recurrence rate was 37.5% [31]. Del Corso et al. had the longest period of follow up (60 months). They compared Nd:YAG laser evaporation and CO_2_ laser excision. Del Corso et al. reported 28.5% recurrence, and no significant difference was found between the two treatment groups. However, CO_2_ laser excision had better results than the Nd:YAG laser evaporation, considering nonhomogeneous OL and OL with mild dysplasia [32]. Another long-term follow-up study (5 years) was performed by Arduino et al. They compared the Er:YAG laser and traditional scalpel surgery. Healing was at the level of 52.99% without significant differences between laser and classical surgery [33].

In a previous study, we used photodynamic therapy (PDT) as a treatment method for leukoplakia, and a complete response was lower than in our present study. After PDT application, a total response was achieved in 29% of lesions, and 59% was a rating of partial response. Lack of treatment effect was seen in 12% [34]. Han et al. also used photodynamic therapy, and they reported 55.2% saw a complete response and 31.0% a partial response [35]. Yao et al. used the combination of PDT and CO_2_ laser. They demonstrated that this is a promising treatment method, especially for a large lesions [36]. Kothe et al. examined images of normal larynx and larynx leukoplakia. They used automatic classification for colour texture analysis, which resulted in 71% for leukoplakia and 97% for normal tissue [37]. Raja et al. used a texture analysis in the case of oral cancers involving the buccal mucosa and assessed its effectiveness in differentiating between various grades of tumours. They revealed that TA on computed tomography images is a promising method in the characterization of oral cancers involving the buccal mucosa [38]. Digital analysis by means of discrete wavelet transformation shows, in places operated by the laser, permanent changes in the appearance of the oral mucosa. These changes concern the macrostructure (large oval fields) as opposed to a fully healed laser-treated microstructure. This is demonstrated by the analysed image texture features derived from the run-length matrix and co-occurrence matrix. This discrepancy in observation may be related to the fact that still there were no leukoplakia-inducing factors removed in the series of patients presented here or knowledge of the metabolic background of pathologies. These factors cannot be removed by local/topical treatment.

Our previous study revealed that fractal dimension of OL foci of the tongue examined during a photodynamic diagnosis procedure was higher than FD during standard white light examination [34]. Our present study revealed no differences between the reference and post-op groups, and no differences were observed between the reference and intermediate groups in contrast to texture analysis.

Lucchese et al. used fractal analysis in the case of complexity of the vascular patterns of oral lichen planus (OLP). They revealed a close relationship between abnormal vascular architecture and atrophic-erosive OLP [39].

Lampros et al. examined the vascular pattern of histologic sections. They reported that carcinomas presented higher mean values of vascular fractal dimension and total vascular area compared to normal mucosa [40]. Rasha et al. examined microscopic slides of mucosa and squamous cell carcinoma (SCC). Fractal dimension increases as the complexity increases from normal to dysplasia and then to SCC [41]. Yang et al. confirmed that fractal dimension provided a precise and theoretically appropriate approximation of cell nuclear structure properties, especially their shape complexity. Fractal dimensions of high-grade dysplasia were significantly lower than those of low-grade dysplasia [42]. In most fractal dimension studies based on microscopic or X-ray image analysis, the authors did not find any similar studies based on intraoral photos. It is important to remember that the oral cavity is not an easy place to take photos due to the irregular shapes, reflections, and softness of some structures (tongue, cheeks), which are sensitive to any tensions. These tensions are able to deform the shape of the examined lesions, and classical methods of measurement fail, so it is important to find methods to bypass these limitations. Our results show that texture and fractal dimension analysis may be helpful in bypassing these problems.

It was found that in the clinical material studied, recurrence of OL after the treatment protocol was ten times more frequent in the tongue or cheek than in the mouth or gingiva. Some primary features of the second order in the appearance of the OL (FDA and DifEntr/LngREmph) seem to have a prognostic significance for the occurrence of a recurrence. However, laser therapy destroys the lesion, and in the intermediate period, all treated sites become similar. The visual characteristics of the mucous membrane after the proposed treatment are similar, and it is not possible to predict on the basis of the proposed photographic image analysis which sites will lead to recurrence. Therefore, the greatest importance should be attached to the assessment of the primary change.

Some studies reveal that a smartphone’s camera may be useful in diagnosis and telemedicine processes [43,44]. Maier et al. applied fractal analysis of pigmented moles using a smartphone’s camera. They had an 8 megapixels matrix, but only 1920 × 1080 resolution was applied during their studies [45]. Breslauer et al. used a cell phone’s camera and an experimental fluorescence microscope for the imaging of tuberculosis and an automated image analysis. They revealed that only 3.2 megapixel (2048 × 1536 pixel, 2.7 μm pixel size) was enough to get a proper resolution and quality of images useful for mathematical analysis [46]. Cameras in telephones nowadays have sensors with a gigantic number of pixels, e.g., 40 million. On the other hand, the authors are familiar with the issue of diffractive limit and relatively low tonal depth. Therefore, in texture analysis 8-bit images were not analysed but reduced to 6-bit. Authors can also postulate that this is the gateway to the widespread use and creation of a mobile-phone-based application for automatic leukoplakia testing.

## 5. Conclusions

Er:YAG laser is an effective treatment method in the case of OL.

Leukoplakia treatment by Er:YAG laser is an effective modality as revealed by the oral mucosa microstructure. Most often in medicine, restitutio ad integrum is not achieved, and in the macrostructure of the photographic image of the treated sites, changes requiring further clinical supervision can be observed.

Texture and fractal dimension analysis are promising methods to estimate the effectiveness of oral leukoplakia treatment, but texture analysis seems to be much more valuable. These two methods may be the basis of a remote diagnosis system of oral lesions in a future study.

## Figures and Tables

**Figure 1 materials-13-03614-f001:**
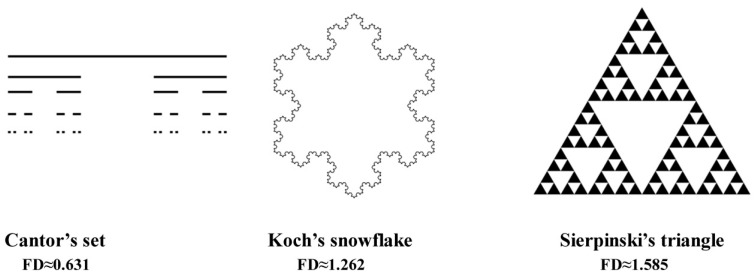
Examples of classical fractal and their fractal dimension value.

**Figure 2 materials-13-03614-f002:**
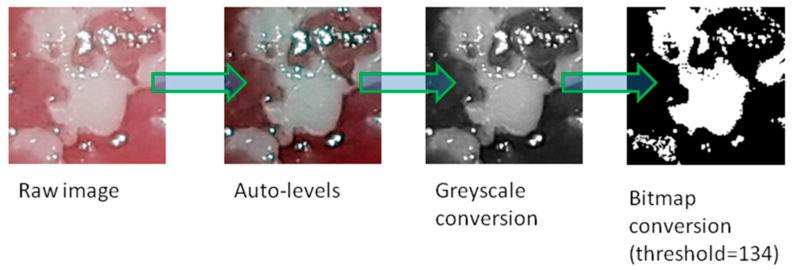
All graphical operations to prepare an image for texture and fractal dimension analysis.

**Figure 3 materials-13-03614-f003:**
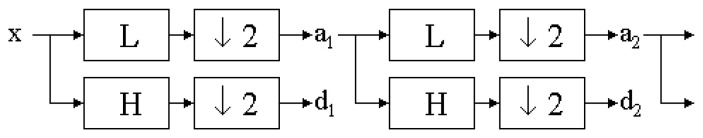
The discrete wavelet transform tree.

**Figure 4 materials-13-03614-f004:**
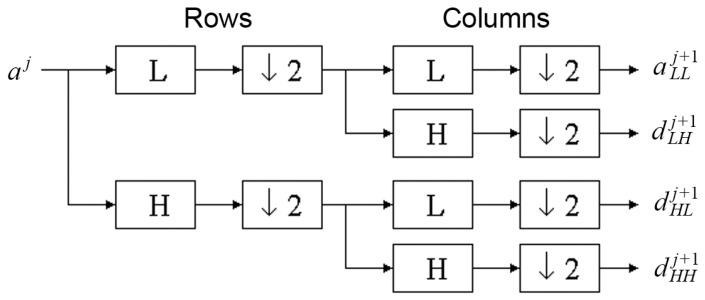
Wavelet decomposition for two-dimensional photography.

**Figure 5 materials-13-03614-f005:**
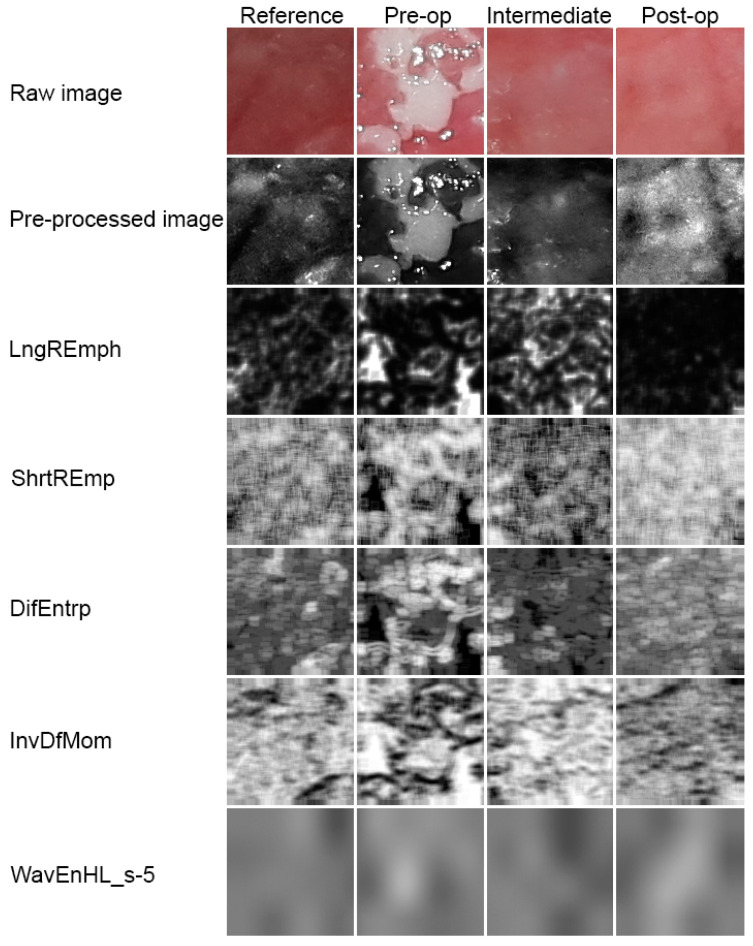
Image processing. Raw photographic files were pre-processed into 8-bit grey level images. Next, five series of texture feature maps are presented.

**Figure 6 materials-13-03614-f006:**
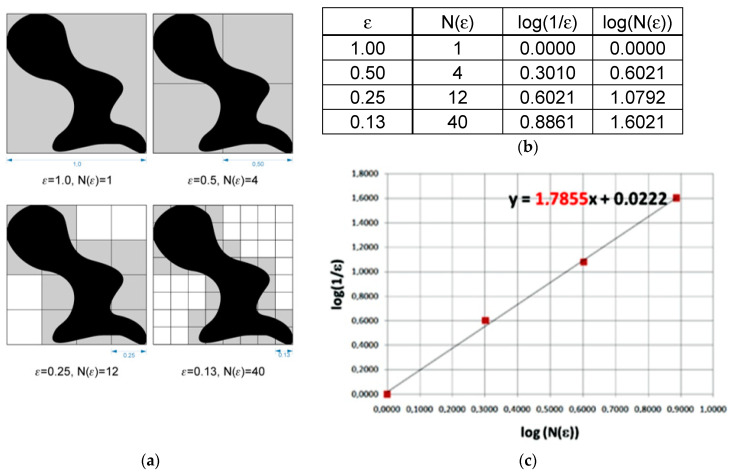
Graphical interpretation of the counting box method for fractal dimension measurement. (**a**)—grids needed to cover the studied shape at the sequential stages, (**b**)—values of log(1/ε) and log(N(ε)) in function of N and ε, where: ε—length of box that creates a mesh covering surface with the examining pattern; N(ε)—minimal number of boxes required to cover the examining pattern, (**c**)—interpretation of the fractal dimension as calculating the value of the directional factor of straight line.

**Figure 7 materials-13-03614-f007:**
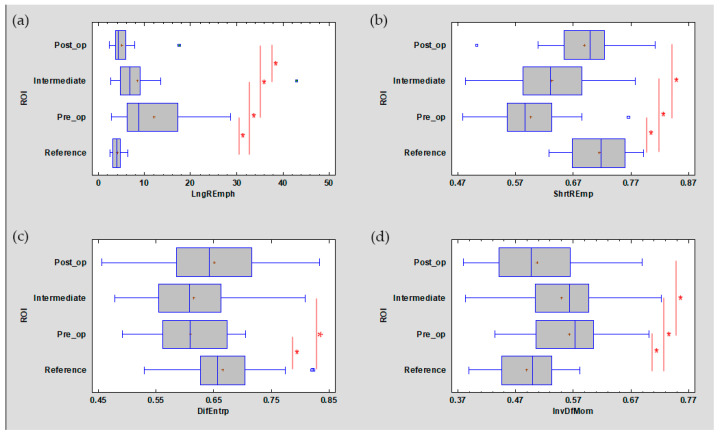
The analysed image texture features derived from the run-length matrix ((**a**) LngREmph, (**b**) ShrtREmp) and co-occurrence matrix ((**c**) DifEntrp, (**d**) InvDfMom). The values of all features presented here are returned after treatment to the value characteristic for the reference mucosal image. + average, blue vertical line indicates median, * significant difference (*p* < 0.05).

**Figure 8 materials-13-03614-f008:**
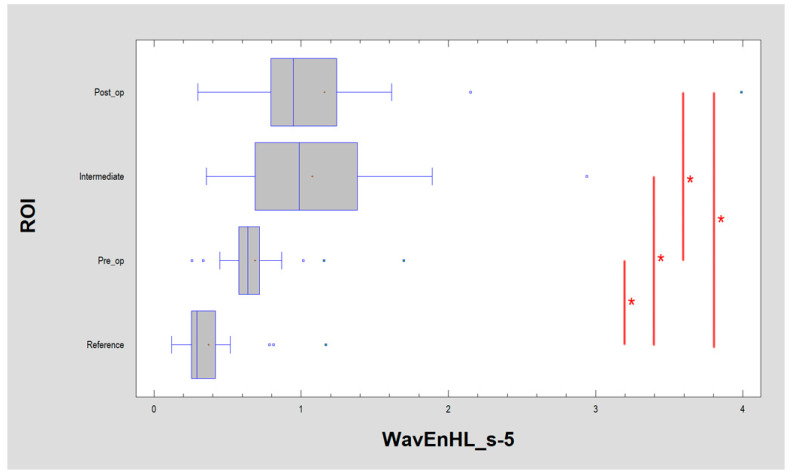
Recognition of large structures in the field of vision (discrete wavelet transform) revealed in the image of the buccal mucosa after surgery, representing a difference from the reference image. These differences consist of the presence of oval fields with a different structure than normal mucous membrane. + average, blue vertical line indicates median, * significant difference (*p* < 0.05).

**Figure 9 materials-13-03614-f009:**
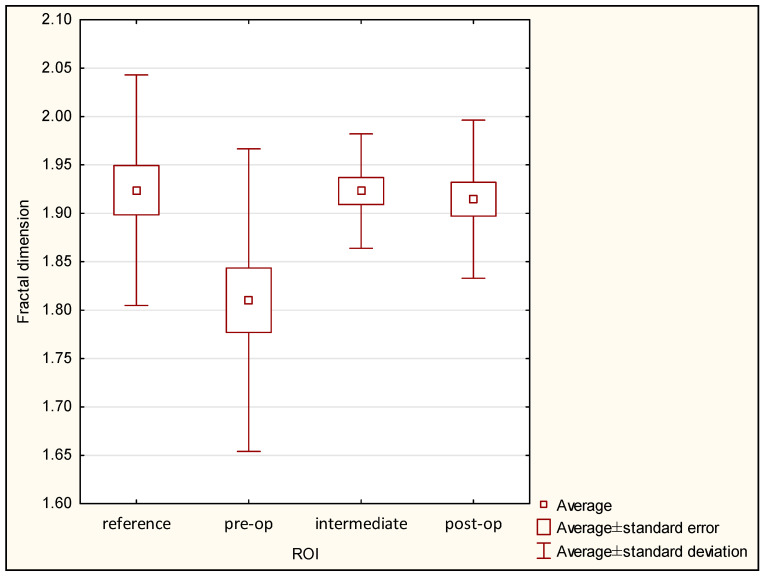
Fractal dimension values in ROI groups.

**Figure 10 materials-13-03614-f010:**
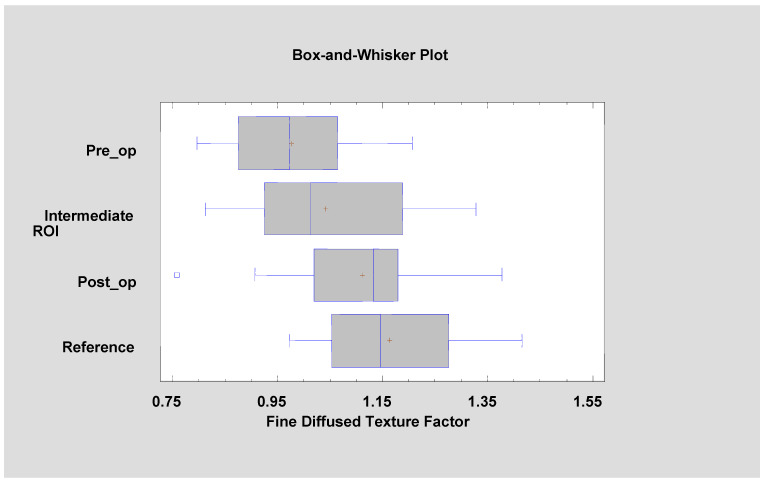
Factor 1 (fine diffused texture factor, FDT factor) calculated for the tested patients. It is noticeable that its value is approaching the mucosal reference value with the progress of leukoplakia treatment (ANOVA, F = 9.38, *p* < 0.001).

**Figure 11 materials-13-03614-f011:**
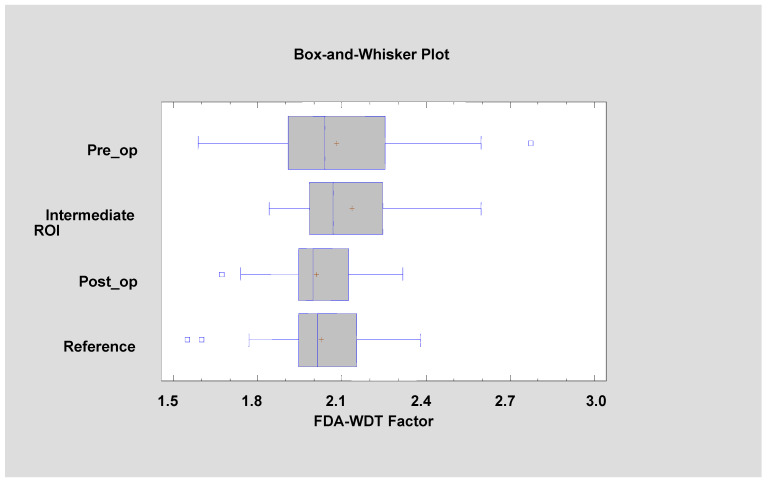
Factor 2 (fractal dimension analysis–wavelet discrete transform factor, FDA–WDT factor) calculated for the tested patients (Kruskal–Wallis test = 4.149, *p* = 0.246).

**Table 1 materials-13-03614-t001:** Digital texture analysis of oral mucosa photographic images after leukoplakia treatment by Er:YAG laser (average ± standard deviations). Abbreviations: ROI—region of interest, i.e., sample of image, KW—Kruskal–Wallis.

Texture Feature	ROI				KW Test
	Reference	Pre-Operational	Intermediate	Post-Operational	
	Mucosa		Period		
LngREmph	4.038 ± 1.171	12.0427 ± 8.503	8.431 ± 7.861	5.071 ± 2.966	*p* < 0.001
ShrtREmp	0.715 ± 0.051	0.596 ± 0.067	0.633 ± 0.079	0.689 ± 0.062	*p* < 0.001
DifEntrp	0.665 ± 0.075	0.610 ± 0.062	0.615 ± 0.087	0.651 ± 0.089	*p* < 0.05
InvDfMom	0.489 ± 0.056	0.563 ± 0.066	0.550 ± 0.080	0.507 ± 0.076	*p* < 0.001
WavEnHL_s-5	0.372 ± 0.241	0.688 ± 0.285	1.077 ± 0.601	1.158 ± 0.711	*p* < 0.001

**Table 2 materials-13-03614-t002:** Results of Kruskal–Wallis one-way analysis of variance for fractal dimension analysis (*p*-values).

ROI				
Versus	Reference	Pre-op	Intermediate	Post-op
reference	-	0.0001	0.3846	0.4659
pre-op	0.0001	-	0.1758	0.0778
intermediate	0.3846	0.1758	-	1.0000
post-op	0.4659	0.0778	1.0000	-

**Table 3 materials-13-03614-t003:** Factor analysis.

Factor	Eigenvalue	Percent of Variance	Cumulative Percentage
Factor 1	2.27223	56.806	56.806
Factor 2	1.00342	25.085	81.891
Factor 3	0.684531	17.113	99.004
Factor 4	0.0398214	0.996	100.000

**Table 4 materials-13-03614-t004:** Factor loading matrix after varimax rotation.

Feature	Factor 1	Factor 2	Estimated	Specific
	FDT Factor	FDA–WDT Factor	Communality	Variance
FDA	0.114858	0.813578	0.675101	0.324899
WavEnHL_s-4	0.167196	0.784505	0.643402	0.356598
DifEntr/LngREmph	0.971237	0.190267	0.979502	0.0204976
ShrtREmp	0.976915	0.152573	0.977642	0.0223583

Abbreviation: FDA: fractal dimension analysis; DifEntr/LngREmph: Difference Entropy, Long-run emphasis inverse moments; ShrtREmp: short-run emphasis inverse moments; FDT: fine diffused texture; FDA–WDT: fractal dimension analysis–wavelet discrete transform.

**Table 5 materials-13-03614-t005:** Selected calculation used for evaluation recurrence rate.

Estimator	Pre-Operational	Intermediate Period	Post-Operational	KW Test
DifEntr/LngREmph	0.08 ± 0.06	0.11 ± 0.07	0.16 ± 0.06	*p* < 0.001
FDT Factor	0.98 ± 0.11	1.04 ± 0.15	1.11 ± 0.13	*p* < 0.001
FDA–WDT Factor	2.08 ± 0.27	2.14 ± 0.20	2.01 ± 0.15	n.s.

Abbreviations: DifEntr/LngREmph: ratio of difference entropy to long-run emphasis inverse moments; FDT factor: fine diffused texture factor; FDA–WDT factor: fractal dimension analysis–wavelet discrete transform factor; KW: Kruskal–Wallis.

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
