# Peer review of "Application of Texture and Fractal Dimension Analysis to Estimate Effectiveness of Oral Leukoplakia Treatment Using an Er:YAG Laser—A Prospective Study"

_materials, 2020, doi:10.3390/ma13163614_

Round 1

Reviewer 1 Report

The paper is very interesting but it would be nice to integrate the bibliography of the introduction considering the latest papers relating to classification of Oral Leukoplakia and to different therapies such as:

Alessandro Villa, Antonio Celentano, Ingrid Glurich, Wenche S. Borgnakke, Siri Beier Jensen, Douglas E. Peterson, Konstantina Delli, David Ojeda, Arjan Vissink, Camile S. Farah World Workshop on Oral Medicine VII: prognostic biomarkers in oral leukoplakia; a systematic review of longitudinal studies : Oral Dis. 2019 June; 25 (Suppl 1): 64–78. doi: 10.1111 / odi. 13087

Giovanni Lodi, Roberto Franchini, Saman Warnakulasuriya, Elena Maria Varoni, Andrea Sardella, Alexander R Kerr, Antonio Carrassi, L CI MacDonald, Helen V Worthington: Interventions for the treatment of oral leukoplakia to prevent oral cancer. Cochrane Oral Health Group Cochrane Database Syst Rev. 2016 Jul; 2016 (7): CD001829. Published online 29 July 2016: doi: 10.1002 / 14651858.CD001829.pub4.

Besides, it would be advisable improve the description of the intervention by indicating at what depth the excision using Erb-Yag laser was performed. This is a very important information for the evaluation of post-treatment recurrence and therefore for a careful analysis of the results.

Author Response

Dear Reviewer,

We added two references up to discission section, which you indicated.

All laser procedures were limited to the epithelial layer of the oral mucosa to decrease risk of scar formation. We add this information to the materials and methods section.

Best regards,

Authors.

Reviewer 2 Report

Generally, this was interesting approach for OL. If we can find the difference between before laser treatment and after laser treatment without TA and FDA, we may have a difficulty in finding any value of these methods. In spite of complex image processing, the findings fro TA and FDA looked not much different to those findings by bare eyes. If TA and FDA can predict the recurrence at the time of immediate after laser treatment, it will be much more interesting. Unfortunately, there was no trials for this. In this study, 34% of lesions were recurred. TA and FDA analysis for those will be much more interesting.

1. In abstract, authors concluded that Erb-Yag laser was an effective treatment method for OL. This conclusion seemed to be based on response rat. However, it was unclear how TA and FDA contributed on estimating the effectiveness of laser treatment. If authors wanted to claim these, the difference in TA and FDA findings between well-responded lesion and poorly responded lesion should be shown. Otherwise, the readers can't determine whether TA and FDA were helpful or not.
2. Authors showed complex steps of image processing and digitised value. As OL is white lesion, it is not suprising that bright pixels were increased in pre-op images. The values of post-op images were similar to those of references. To say differently, post-operative oral mucosa looks like references, but different to pre-op mucosa. This makes a sense. Unfortunately, this image processing can't predict the recurrence among post-op lesion. 

Author Response

Dear Reviewer,

  1. We added some statistical analysis to estimate an risk of recurrences and prognostic value. It was found that in the clinical material studied, recurrence of OL after the treatment protocol was ten times more frequent in the tongue or cheek than in the mouth or gingiva. Some primary features of the second order in the appearance of the OL (FDA and DifEntr/LngREmph) seem to have a prognostic significance for the occurrence of a recurrence. However, laser therapy destroys the lesion, and in the intermediate period all treated sites become similar. The visual characteristics of the mucous membrane after the proposed treatment are similar, and it is not possible to predict on the basis of the proposed photographic image analysis which sites will lead to recurrence. Therefore, the greatest importance should be attached to the assessment of the primary change.
  2. All images processing are necessary to obtain source material to analysis. Auto-levels tool enables stretching a histogram which increases contrast between lesion and background. Grey scale conversion is needed to creating bitmap. Classical counting box method (fractal dimension analysis) requires bitmap format (one bit notation – 1 – signal, 0 – background). Threshold of this conversion (134) was experimentally chosen to achieve sharp borders of lesions, threshold level of 128 (default binary conversion) offered much blurred these borders. Texture analysis requires grey-scale conversion.

Best regards,

Authors.

Reviewer 3 Report

Dear Authors,

I read the manuscript carefully. Overall, your idea is interesting but some critical issues have emerged that do not allow, in my idea, to allow the publication of the manuscript. Considering that the proposed analysis is based on the acquired images, I do not consider the quality and repeatability of the photos acquired via a smartphone sufficient. It would have been ideal to use a reflex camera with a lens dedicated to intraoral photos (macro lenses between 80 and 110 mm) with ring flash or scorpion-like system.

Unfortunately, this choice negatively affects all the work presented, even if well written. In any case, I refer to the decision of the editorial board and to the reports of the other reviewers.

Author Response

Dear Reviewer,

Cell phone camera was used because in the future study we will try to create a system of remote diagnosis of oral mucosa lesions on the base of cell phone photography to simplify whole process of dentist’s side. In our opinion 12 megapixel (1/2.55” sensor size and 1.4㎛ pixel size) with focus distance 10cm offers optimal optical resolution.  In our previously study we used Canon 500d camera, Canon 50mm lens, ring flash-light and intermediate ring to reduce distance of focus. Images were analyzed in case of leukoplakia and lichen planus by fractal dimension analysis and quality of digital material was similar to material from modern cell phone camera.

Best regards,

Authors.

Reviewer 4 Report

Dear authors, 
I appreciate the efforts of the manuscript with the title “Application of texture and fractal dimension analysis to estimate effectiveness of oral leukoplakia treatment using Erb:YAG laser
”.

-Why did you preferred an Erb:yag over the other laser devices?

-Line 98 how exactly were the specimen taken

-Is it a limitation to your method if the OL is large and homogenic, so no changing image structures?

- You performed a histopathological baseline examination. Were there any correlations between the initial dysplasia grade and the effectiveness of the laser application?

-Why were smokers not excluded? As you describe later in the discussion smoking has a potential effect on the outcome. What about alcohol or any other diseases.

Author Response

Dear Reviewer,

  1. The Erb:YAG laser emits light with a 2940nm wavelength which corresponds to the main peak of water absorption. Most of the energy is absorbed in the epidermis and papillary dermis. It is more superficial ablation with less thermal damage. We try to operate as superficial as possible to avoid risk of scar formation.
  2. After local anaesthesia specimen of lesion was taking out for histopathological examination. Excision site was chosen on the border between healthy mucosa and lesion. In case of widespread lesions two or three samples were collected from most representative sites. Histopatholocal slides were haematoxylin and eosin stained. This description was added to the materials and method section.
  3. The main limitation of mathematical images analysis is repeatability of taking intraoral photos especially light reflexes on wet mucous membrane. Hard accessible sites of oral cavity are the other limitation this method due to difficulties of taking photos. All photos were taken perpendicular to mucous membrane to achieve repeatability. Focus point was locked at the same distance to achieve the same scale of photos.
  4. Dysplasia occurrence was an excluding criteria. It was added to the materials and methods chapter.
  5. Smoking of cigarettes were another excluding criteria (it was added to the materials and methods chapter). All patients filled up questionnaire about eating habits, for example: alcohol or spicy foods consumption. Nobody signed alcohol consumption in other way than occasionally.

Best regards,

Authors.

Reviewer 5 Report

The authors conducted a study aimed to evaluate the effectiveness of oral leukoplakia treatment using Erb:YAG laser by using texture and fractal dimension analysis. This is a very interesting theme and the authors have a large experience in this field.    The manuscript is well written but there are some minors issues to improve:  

1- The design of the study is unclear whether prospective or retrospective. Please include in the title the type of study design.

2- The plagiarism test shows 35% similarities to other publication (see the attached pdf), almost the material and methods part and other paragraphs are a self plagiarism from "Jurczyszyn, K., Kazubowska, K., Kubasiewicz-Ross, P., Ziółkowski, P., & Dominiak, M. (2018). Application of fractal dimension analysis and photodynamic diagnosis in the case of differentiation between lichen planus and leukoplakia: A preliminary study. Advances in Clinical and Experimental Medicine, 27(12), 1729-1736." and " Jurczyszyn, K., Kubasiewicz-Ross, P., Nawrot-Hadzik, I., Gedrange, T., Dominiak, M., & Hadzik, J. (2018). Fractal dimension analysis a supplementary mathematical method for bone defect regeneration measurement. Annals of Anatomy-Anatomischer Anzeiger, 219, 83-88."

This part have to rephrased to avoid Text recycling.

"Text recycling: If the author uses large portions of his/her own already published text in his/her new manuscript, it is called text recycling. It can be detected by plagiarism software. It can be handled as per the COPE guidelines." Dhammi, I. K., & Ul Haq, R. (2016). What is plagiarism and how to avoid it?. Indian journal of orthopaedics, 50(6), 581–583. https://doi.org/10.4103/0019-5413.193485

Author Response

Dear Reviewer,

  1. We changed the title to: Application of texture and fractal dimension analysis to estimate effectiveness of oral leukoplakia treatment using Erb:YAG laser – a prospective study.
  2. We rephrased section (introduction, materials and methods and discussion) to avoid text recycling.

Best regards,

Authors.

Round 2

Reviewer 2 Report

Authors addressed the revised version in response to the reviewer's critics. Considering the characteristics of OL, the prediction of recurrence with authors' method would be impossible. However, authors' method is helpful for the assessment of therapeutic result as authors stated. 

Author Response

Dear Reviewer,

Yes, the authors share the reviewer's opinion. The presented algorithms allow to diagnose OL and, on the basis of primary features of the lesion, to predict resumes. However, the use of laser causes complete evaporation of white plates - there is nothing left for visual evaluation either in the intermediate period or after the treatment. The authors believe that laser is a good tool for OL treatment.

Best regards,

Authors.

Reviewer 3 Report

Dear Authors,

I understand your reasons but the photos attached to the manuscript are of very bad quality. High quality photos would still be needed even if taken with a smartphone camera. However, the purpose of the work and the idea of developing an oral telemedicine application do not seem related to me. Therefore, I am very sorry to confirm my decision on this manuscript.

Author Response

Dear Reviewer,

Cameras in telephones nowadays have sensors with a gigantic number of pixels, e.g. 40 million. On the other hand, we are familiar with the issue of diffractive limit and small tonal depth. Therefore, in texture analysis we don't try to analyze 8-bit images, but reduced to 6-bit. We can also postulate that this is the gateway to the widespread use and creation of a mobile phone-based application for automatic leukoplakia testing. Even if we use traditional lens the optical resolution is determined by the les diameter. When we closed shutter of lens to achieve deeper focus plane we decrease optical resolution by reducing active diameter of lens. In case of shorter focus length of cell phones lenses, focus depth is larger and it is possible to achieve sharp image in more open shutter.

Some studies revealed that the smartphone’s camera may be useful in diagnosis and telemedicine process. Maier et al. applied fractal analysis of  a pigmented moles using smartphone’s camera. They had 8 mega pixels matrix but only 1920x1080 resolution was applied during their studies. Breslauer  et al used cell  phone’s camera end experimental fluorescence microscope to imaging of tuberculosis and automated image analysis. They revealed that only 3.2 megapixel (2048x1536 pixel, 2.7um pixel size) is enough to get a proper resolution and quality of images useful for mathematical analysis. Cameras in telephones nowadays have sensors with a gigantic number of pixels, e.g. 40 million. On the other hand, we are familiar with the issue of diffractive limit and small tonal depth. Therefore, in texture analysis we don't try to analyze 8-bit images, but reduced to 6-bit. We can also postulate that this is the gateway to the widespread use and creation of a mobile phone-based application for automatic leukoplakia testing.

Thank you for your opinion. In the next study we will perform both camera and cell phone camera photos to estimate possible differences in a texture and a fractal dimension analysis between these two source of images.

Best regards,

Authors.

Reviewer 4 Report

Taking into account the changes made to the text during the revision, the publication is now acceptable.

Author Response

Dear Reviewer,

Thank you for your decision.

Best regards,

Authors.

Reviewer 5 Report

Thank you for the revision, all requested changes were done

Author Response

(The authors gave the same response as above.)
